# Questioning ‘Informed Choice’ in Medical Screening: The Role of Neoliberal Rhetoric, Culture, and Social Context

**DOI:** 10.3390/healthcare11091230

**Published:** 2023-04-26

**Authors:** Emma Grundtvig Gram, Alexandra Brandt Ryborg Jønsson, John Brandt Brodersen, Christina Sadolin Damhus

**Affiliations:** 1Center of General Practice, Department of Public Health, University of Copenhagen, 1353 Copenhagen, Denmark; 2Primary Health Care Research Unit, 4100 Region Zealand, Denmark; 3Department of People and Technology, Roskilde University, 4000 Roskilde, Denmark; 4The Research Unit for General Practice, Department of Social Medicine, University of Tromsø, 9019 Tromsø, Norway

**Keywords:** informed choice, medical screening, decision-making, screening participation, biopower, governmentality

## Abstract

Participation in medical screening programs is presented as a voluntary decision that should be based on an informed choice. An informed choice is often emphasized to rely on three assumptions: (1) the decision-maker has available information about the benefits and harms, (2) the decision-maker can understand and interpret this information, and (3) the decision-maker can relate this information to personal values and preferences. In this article, we empirically challenge the concept of informed choice in the context of medical screening. We use document analysis to analyze and build upon findings and interpretations from previously published articles on participation in screening. We find that citizens do not receive neutral or balanced information about benefits and harms, yet are exposed to manipulative framing effects. The citizens have high expectations about the benefits of screening, and therefore experience cognitive strains when informed about the harm. We demonstrate that decisions about screening participation are informed by neoliberal arguments of personal responsibility and cultural healthism, and thus cannot be regarded as decisions based on individual values and preferences independently of context. We argue that the concept of informed choice serves as a power technology for people to govern themselves and can be considered an implicit verification of biopower.

## 1. Introduction

The purpose of medical screening is the early detection of pathological changes in seemingly healthy people, and the identification of those at risk of having or developing a given disease, to attain a better prognosis [1,2,3]. While potentially reduced morbidity and mortality are the benefits of screening, all screening programs cause unintended harm. Therefore, most countries have protocols to evaluate screening programs before their eventual implementation. Following this, Denmark holds ten criteria for the implementation of medical screening. One of the criteria is that the invitation to participate in screening should include understandable, comprehensive, and nuanced information while emphasizing participation as voluntary [4]. Likewise, The Danish Health Act requires that any medical decision should be based on informed consent defined as based on best-available evidence communicated in an understandable language that is tailored to the individual or target group [5]. Such institutional regulations focus on the right to information and rely on the capacity of the citizens to make an informed choice based on that information. Academic definitions of the informed choice in medical screening vary, yet share some common features which can be summarized into three ‘assumptions’: (1) that balanced and neutral information is available, (2) that the decision-maker can understand and interpret the provided information, and (3) that the decision-maker can relate the information to personal values and preferences (Table 1) [6,7,8,9].

However, societal tendencies and governmental interests threaten the informed choice in medical screening. For example, screening might only have a financial benefit if taken up by the majority of the population. Furthermore, reducing the absolute risk of dying from the disease screened for requires that a predominance of the invitees participates. The participation rate is, therefore, used as an institutional quality indicator in screening programs [10,11,12]. In Denmark, the Danish Health Authority encourages participation in governmental information leaflets and pre-books appointments for mammography screening to maximize participation, and thereby increase the effect of the program [13,14,15,16,17,18]. Further, the Danish Health Authority collaborates with patient organizations such as the Danish Cancer Society regarding campaigns aimed to maximize participation. At the same time, the authorities are obligated by law to practice voluntary and informed decision-making.

Yet, despite research on aspects of informed choice in medical screening, there is a lack of knowledge on how the concept and the assumptions it relies on are informed by and translated into the cultural and social context. This calls for scrutinizing the assumptions of informed choice, and analyzing whether these preconditions meet the principles of voluntary and informed participation. In this article, we build upon prior efforts to analyze perceptions and experiences of, and contextual conditions affecting participation in, screening programs, to discuss informed choice as a phenomenon based on the three abovementioned assumptions.

## 2. Materials and Methods

To investigate the rationales and understandings of informed choice in medical screening, we will apply the three assumptions as an analytical lens on our research and selected published work that relates to decision-making, participation, and informed choice in medical screening.

Denmark offers equal and free access to health care, and therefore is an optimal vantage point from which to study informed choice in medical screening.

For the analysis, we do an audit of primary sources by extracting data from qualitative articles on participation in medical screening and do a document analysis [19,20,21]. We include articles that relate to medical screening, informed choice, patient involvement, or participation in medical initiatives. Thus, we include articles that relate to medical screening, informed choice, patient involvement, or participation in medical initiatives. We purposefully examine qualitative studies focusing on these subjects and due to the paucity of such, we do not exclude articles in which the author group is involved (See Table 2) [19]. This empirical data form the base for the current analysis.

We conduct the document analysis based on the selected qualitative research articles. By this method, we evaluate the articles, interpret their findings, and analyze them to gain a new understanding of their meaning. In this analysis, we condense the information about participating in screening and develop the findings and interpretations the articles provide. This analysis is based on two sub-questions relating to the aim of this article: (1) how is ‘informed choice’ framed and presented regarding participation in medical screening, and (2) what priorities, attitudes, and perspectives of patients/citizens can be identified about participating in medical screening?

We extract data from the selected articles, and discuss the findings of the articles and how the findings relate to the two sub-questions (Table 2). We do these discussions within the author group. We analyze the data using meaning condensation, in which we convert the essence of particular findings and quotes into descriptive sentences [20,21]. We then do a thematic coding and sort each extract of data into one of the three assumptions about the informed choice. We provide illustrative quotes in the analysis of each assumption to stress important points.

### Theoretical Onset

In Denmark, as in most other similar countries, notions of neoliberalism can be traced throughout most of society [29,30]. In regard to health, however, neoliberalism is not to be understood as the classic economic theory but rather what Foucault coined as a form of governmentality. Such a neoliberal paradigm in health places emphasis on disease prevention attained through individual responsibilities, which then forms the moral belief that individuals are responsible for their health [31,32]. In this paradigm, modes of citizenship are negotiated and accentuated at the expense of others’ actions, and knowledge about lifestyle risks internalizes ideas about ‘proper health behavior’. Each citizen is considered responsible for their health. Bio-medicalization theory describes how the body is no longer perceived through dichotomies, such as normal or abnormal, but as something that can be transformed and manipulated for the better [33]. This can be related to Foucault’s concept of biopower as a form of biomedical governance that seeks not only to control people but for people to control themselves through the shaping of truths, alterations of subjectivities, and patient enablement [34]. This kind of power of the medical practice causes people to take up “(…) a new kind of ‘clinical life’” [34] that make individuals feel ethically obliged and willing to promote their health and life expectancy by transforming their bodies with biomedical intervention [33,35]. Bio-medicalization theory argues that people are given more responsibility and moral obligations to maintain health, with a constant focus on risk factors and the ability to transform bodies, and that people come to rely on the healthcare institution to fulfill this responsibility.

Applying this theoretical lens to our analysis, we put forth some of the underlying, and often implicit, understandings and moral modalities that nonetheless compile a context in which informed choices are to be experienced, negotiated, and acted upon.

## 3. Results and Analysis

Table 1 gives an overview of the included articles, how they were analyzed and coded in regard to the two sub-questions, and how these support our overall analysis and interpretation.

### 3.1. Assumption 1: The Availability of Balanced and Neutral Information

The first precondition for making an informed choice is the availability of balanced and neutral information on which the potential participant can rest their decision [18]. While most medical screening programs claim to provide such information in decision aids or pamphlets, research has established that the information is not balanced; it understates or omits the harms, yet overstates the benefits [9,14,16,22,36,37,38,39,40,41,42]. Information material that does not present the benefits and harms equally cannot be considered balanced or neutral, and cannot support informed decision-making.

This is, for example, seen in a study that investigates information pamphlets for women invited to a cervical cancer screening. The authors scrutinize pamphlets from 12 countries and find that none of these provide adequate information about the benefits and harms of screening; the pamphlets contain only a median of four out of 23 information items (17.4%) [9]. On average, one in three pamphlets mention the potential harms, such as false positives or overdiagnosis, yet only one in 12 (8.3%) provides quantifications of these, such as frequencies or relative risk estimates [9].

Another concern is how the information is framed and the use of nudging strategies to increase participation [22]. A nudge is any factor that alters people’s behavior without forbidding any options [43]. A systematic review identifies six categories of framing effects including nudges used in cancer screening pamphlets: misleading presentation of statistics, misrepresentation of harms vs. benefits, opt-out systems, fear appeals, and recommendation to participate [44]. A Danish randomized controlled trial (RCT) shows that people are more likely to participate in screening if information pamphlets with the screening invitation contain framing effects, and that willingness to participate increases proportionally with the number of framing effects incorporated in the invitation [40]. One of the framing effects that increases participation is a governmental recommendation to participate. According to Thaler and Sunstein, a nudge has to be identifiable to be a nudge; if not, it is classified as manipulative [45]. Nonetheless, the majority of the RCT participants are not able to identify the framing effects [45]. This hints that some of the framing effects that are used to maximize participation are not nudging but manipulative strategies.

Pre-booked appointments significantly maximize participation in Norway [46], and an interview study conducted among Norwegian women invited to mammography screening, finds that the women consider the pre-booked appointment as if a decision has already been made for them [23]. Framing effects such as pre-booked appointments rely not on individuals to make a choice but instead comply with the governmental aim of maximum participation. The cognitive theory claims that people have a default bias that causes them to follow default options, and therefore screening invitees invited with pre-booked appointments are more likely to participate simply as a default mechanism rather than as informed decision-making [47,48]. This is in discordance with the Danish Health Act, which states that health decisions should be based on informed consent.

Combining this evidence, screening invitations and information material do not seem to comprehensively present all possible benefits and harms, yet contain manipulative framing effects. Thus, they cannot function as a prerequisite for making an informed choice.

### 3.2. Assumption 2: The Decision-Maker Can Understand and Interpret the Information Provided

The second precondition for informed choice is that the decision-maker is capable of understanding and interpreting the information. Research has discussed this capability in terms of health literacy, understood as cognitive and social skills for understanding and using health information [49]. We wish to compile data to underline a different argument; comprehending information is not necessarily linked to intellectual capacities but rather to how the information is presented and incorporated by the individual.

The communication of health statistics is challenging, and even when efforts are made, such as the presentation of natural frequencies, comparable denominators, and pictograms, individuals are challenged in understanding such health statistics [8,50,51,52,53,54]. Communicating the benefits and harms of medical screening is not an easy task [16,55]. When interviewing women about the national pamphlets for breast and cervical cancer screening in Denmark, we notice that they struggle to understand the health statistics presented in these pamphlets [24,25]. However, when we present the statistical information using pictograms, natural frequencies, and comparable denominators, more women can understand the information. However, the women disregard health statistics that conflict with their previous understanding of the screening program. Using cognitive dissonance theory, we suggest that dissonance commends between the written information and the women’s interpretation of the screening program, which we define as a ‘perception gap’ [24,25,56]. The women tend to downplay the harms, either by adding information that makes the harms less relevant or by ignoring them in their explanations. For example, one woman named Olivia is asked to explain the fact that 86 out of 99 cone biopsies following cervical cancer screening are overtreated [25]:


*“Yes, it’s not sure that you have cancer when you get this test, and maybe it’s unnecessary; but of course, you need to do it anyway. So actually it’s lovely that, out of all the women getting it done, only a few have cancer.”*
(Olivia, 53 years) [25], page 12.

Another woman responds to the fact that there is no difference in length of life when comparing screening participants and non-participants [24]:


*“It doesn’t matter if I don’t live longer, as long as I get saved from dying [from breast cancer].”*
(Mary) [24], page 6.

In an interview study about continuing mammography screening above the recommended age limit, the women reject the risk of harm by self-exempting themselves from the risk of being harmed or downplaying their reaction to or the importance of the harm [26]. In a deliberative poll study, the women do not only disregard information discordant with their previous beliefs but alter their preferences for mammography screening to fit the information available [27].

The perception gap might emerge due to positive presumptions and experiences with the screening programs [57]. According to cognitive bias theory, such positive attitudes are difficult and uncomfortable to change, and consequently, citizens invoke particular interpretations to make the information compatible with existing understandings of screening. Furthermore, people are inclined to base their decisions upon emotions: if you like something you are more inclined to believe in its benefits and that the harms are negligible [58,59]. The literature supports that people have strong positive presumptions about the benefits of screening and generally overestimate the benefits [57,60,61].

Combining this evidence, it seems that the capacity to comprehend information about screening relies on how the information is presented, yet cognitive strains favor screening and alter preferences for screening.

### 3.3. Assumption 3: The Decision-Maker Can Relate Information to Personal Values and Preferences

The third and last assumption about informed choice is that it requires that the decision-maker can relate the information provided to personal individual aspects such as values and preferences. Research shows that healthcare decisions that incorporate the individuals’ personal values and preferences have better outcomes, but our results point to the challenge of relating personal values and preferences to medical interventions when offered by the authorities. This may be because the social, cultural, and societal contexts influence what is deemed a ‘choice’; nevertheless, moral and social modalities affect that choice [62,63].

First, personal preferences and values are permeated by implicit cultural morals. For instance, in studies on mammography and cervical cancer screening, the participating women regard participating as a moral obligation; they argue that participation is for the greater good of society, including societal finances [22,24]. Some women also express feelings of guilt and apply condescending terms such as lazy to themselves when they have not done what they considered enough to take care of themselves, for example, examining their breasts in between screenings [22,26].

Second, women assign positive personal attributes to participation and identify as responsible citizens who take good care of themselves [22,26,28]. To be offered and recommended preventive health services such as screening shape norms of responsible health behavior and alter what the women consider a ‘good citizen’. Therefore, they are not reluctant to participate in screening, as Charlotte exemplifies [28]:


*“That’s just the way I am. I do what I have to do and what I can do in relation to my body and my health. So, that’s just what you do. When you receive that letter, you just do it.”*
(Charlotte, 57) [28], page 706.

As opposed to doing what is considered the ‘right’ thing and participating, not participating can be accompanied by the fear of regret. In a paradigm, where health is regarded as a super value and health an individual responsibility, fear is a punishment that we administer to ourselves [32]. Regret is one of the counterfactual emotions that are triggered by the availability of alternatives to reality. Regret and blame are evoked by comparison to a norm. This is exemplified in a telephone interview study, where participants judge women to be irresponsible if they do not participate in screening [64]. This fear of regret and being considered irresponsible favors risk-averse choices: participation in screening [48]. This has been presented as the paradox of control because any obtainment of control is accompanied by the fear of its loss [26,32]. Personal preferences and values of screening are likely to be considered irresponsible if they do not comply with the internalized norms of proper health behavior, the norm to attend the screening [65]. However, preferences and values are as likely to be constituted by these norms and the neoliberal discourse of individual responsibility.

Third, personal values and preferences are not formed in isolation but are socially constructed and negotiated. In interviews with women about their decision-making in mammography screening, the women’s decision-making processes are dominated by the attitudes of their family and friends, and the information in the invitation leaflet has little influence [24]. One woman, Lisa talks about her decision-making process:


*“My Mum has also gone to do it [mammography screening], so I’ve been talking to her about it and she is comfortable with it too and thinks it’s worthwhile.”*
(Lisa) [24], page 5.

Individuals have an innate tendency to construct and believe coherent narratives of the past [66]. When individuals evaluate participation in screening, the narratives will favor screening: Did they find a cancer? Well, good thing I went to screening. Did they not find a cancer? Well, good thing I went to screening. It is a powerful intuition that, for example, because my mom went to screening is the reason why my mom is not sick [61].

And finally, in most countries in the Global North, health cultures are shaped in concert with neoliberal paradigms of individual responsibility. In interviews with women about mammography screening, the women express that they regard healthcare services as representative of the welfare society, which takes care of its citizens’ well-being [26,28]. The screening was not considered a choice but an exclusive offer that was not to be taken for granted. One of the women, Anne expresses her gratitude for the screening program [28]:


*“I felt that it was amazing; that one could be so lucky as to be invited to screening. I think it’s very generous to be offered such an examination free of charge.”*
(Anne, 70) [28], page 705.

In three studies, the citizens accepted invitations to screening simply because they thought it must be good since it was offered and recommended by the authorities [22,26,28]. For example, a woman named Barbara had participated in mammography screening for several years and states that:


*“I think that when you’re offered screening, then you should take it. Most certainly. A lot of women don’t want to. I don’t get that. We should be pleased that we are offered screening.”*
(Barbara, 72) [26].

This point to how public preventive health services are a symbol that society and the health authorities care for its citizens and that there is no reason to doubt the intentions of the state [23]. These examples show how governmental recommendations and campaigns influence screening decisions in populations or groups that have high trust in authorities. They also show that the decision to participate in screening is affected by societal norms and notions of individual responsibility. Hence, personal values and preferences are inadequate as sufficient conditions for medical decisions.

## 4. Discussion

The information provided to screening invitees does not reflect balanced information about benefits and harms. Further, the information contains manipulative framing effects and therefore cannot be considered neutral. Individuals may access comprehensive information, yet health statistics are often not presented in a comprehensible way. On top of that, individuals may experience cognitive strains in the understanding of the benefits and harms, such as a perception gap. We argue that this was not due to lack of intellectual capacities, but human nature. Last, in terms of making an informed choice, citizens are not able to isolate the choice in regard to personal values and preferences from the cultural and societal context, which induces people to take good care of themselves including participating in medical screenings. As a consequence, citizens will not regard participation as a choice but as a moral obligation. Additionally, the dreadful fear that the decision not to participate in screening will be regretted in the future, for example, due to disease, is an emotion difficult to ignore in the neoliberal decision context. Based on this, the right to informed choice in medical screening is to be regarded merely as a theoretical concept rather than an empirical phenomenon.

### 4.1. Informed Consent as a Modern Power Technology

Neoliberalism in health might enhance patient autonomy and rights, yet presuppose that individuals will be able to make the right decision for themselves when informed properly about the benefits and harms. We have shown that this premise is not as straightforward as assumed by the Danish Health Act and the ethical consensus on medical screening.

The neoliberal paradigm places the responsibility at the individual level contrary to the governmental level. Governments might tolerate more harm when implementing screening because the neoliberal discourse depicts that any individual will make an informed decision to participate. In other words, people can simply choose not to participate if they judge that the harm outweigh the benefits. According to the arguments we have presented, governments and health authorities might accept more harm at a false premise.

We argue that using informed consent as a benchmark for implementing harmful medical practices is an implicit way of verifying a biopower. The informed choice is being used as a modern technology of power for people to control themselves: governmentality. Foucault argues that any modern technology must include elements of freedom and appeal to people’s wants and needs to be considered a power technology [34,67,68]. First of all, citizens are given the right to voluntary and informed choices in medical interventions. This can be regarded as the element of freedom by which it becomes a governmentality strategy, where people are controlled to control themselves. However, although people can voluntarily choose not to undergo screening, they are faced with the fear of regret that follows opposing the norm and being considered irresponsible. In other words, societal and cultural norms about responsible health behavior indirectly disqualify the element of freedom in informed choices about screening. Second, the informed choice in screening invitation appeals to individuals’ wants and needs; the invitations to screening accompanied by governmental recommendations appeal to the individual responsibility to take care of their health. We also argue that framing effects in invitation leaflets contain manipulative elements and that these effects in fact maximize screening uptake. This can be considered as if the governmental appeal alters the subject and the subject identity that as a further consequence also alters actions, what Foucault coins as docile bodies [68]. We show that screening invitees also alter their values and preferences to fit with the information about screening. The biopower thereby masks itself as informed and voluntary choices and forms docile bodies [34,69].

At the center of the practice of biopower and governmentality lies the supposition that the organizational and institutional powers managing this know what is good and proper behavior for individuals or groups of individuals. Thereby, implementing screening with incorporated manipulative elements, recommendations, and appeals to individual responsibility to maximize screening uptake violate informed choice. It also presupposes that what is best for the individual is fixed and in accordance with governmental and public interests. Ideology, notions of individual responsibility, and governmental interests are given the potency to outweigh the scientific evidence about the benefits and harms of screening.

Research shows that this is also present in other health care system and countries [70,71].

### 4.2. Strengths and Limitations

There are some limitations to our simplification of the informed choice. The three assumptions might not be distinctive and dividing these three might not always be meaningful when discussing informed choice in a medical context. Evidence and information may also be evaluated differently by researchers with diverse methodologies or epistemic approaches [72,73].

While showing that the informed choice as a concept is questionable in a screening context, it might not be possible to decide in a specific situation whether or not a citizen has made an informed choice about screening or not. For example, in a study by Lindholdt and colleagues on Abdominal Aorta Aneurism-screening in Denmark, the authors argue that informed individuals are able to reflect on the participation decision if given a choice [74,75]. However, in the study, the choice is not between screening or no screening, but between different versions of a screening program and only people who agree to participate in screening were invited to participate in the study.

Nordic welfare states offer free access to healthcare and serve as a vantage point for studying the informed choice in screening. In countries where screening is paid for privately or through insurances, the role of culture and social context might be less pronounced.

This study builds on recent studies published in well-renowned journals. Yet, we did not perform systematic searches and therefore this paper should not be considered a review representative of the entire field of literature on informed choice in screening. Despite controversies and conceptual divisions within the phenomena of ‘informed choice’ in medical screening, we find that the collected data and our analysis raise an important issue in studying and discussing informed choice and participation in screening. Yet, this paper’s core strength is the interdisciplinarity in which we combine medical knowledge, public health scholarship, and social science theories to give a more nuanced contribution to current conversations on informed choice.

## 5. Conclusions

Governments and health law present participation in medical screening as a voluntary and informed choice; yet, we have shown that this might not be as simple as is assumed. Governments provide screening invitees with incomplete, inaccurate, and unbalanced information. The neoliberal health culture probes screening participation as a moral obligation and associates non-participation with the fear of regret. These aspects hinder the informed choice in screening. Governments and health authorities therefore accept more harm at a false premise if the informed choice is used as a benchmark for the magnitude of harm that can be accepted when implementing screening programs. We argue that the concept of informed choice serves as a power technology for people to govern themselves and an implicit verification of biopower in health. Lastly, using neoliberal arguments of individual responsibility and patient autonomy in health might divert attention from, for example, the socioeconomic and genetic factors from which poor health originates.

## Figures and Tables

**Table 1 healthcare-11-01230-t001:** Three common features of the informed choice.

(1) The availability of balanced and neutral information The decision-maker needs neutral and balanced information on both the benefits and the harms. This means that the decision-maker can access the best-available evidence about the consequences of all of the possible choice options, and that this evidence is presented in a neutral and balanced way.
(2) The decision-maker can understand and interpret the information provided The decision-maker should be able to understand and interpret the information. This means that the information and health statistics should be communicated in an understandable way, and the benefits and harms should be comparable.
(3) The decision-maker can relate information to personal values and preferences The decision-maker should be able to relate the information to their own personal values and preferences. In a screening setting, this would imply that the individual is able to predict and accept the future consequences of their choice.

**Table 2 healthcare-11-01230-t002:** Included articles, sub-questions (SQ), and analysis.

Included Articles	Study Description	SQ1: Framing of ‘Informed Choice’	SQ2: Priorities, Attitudes, and Perspectives of Citizens	Support for Analysis and Interpretation
Damhus et al. 2018 [22]	Interview study on the understandability of information in colorectal cancer screening in Denmark	There is an international consensus that participation in screening should be based on informed consent	Trust in authorities, preventing cancer is important, reassurance of screening, harms are not important, focus on the benefits	Assumption 1: Information is not balanced, understates or omits the harm, yet overstates the benefits. Assumption 3: Participation is a moral obligation.
Østerlie et al. 2008 [23]	Focus groups about the decision-making process among women invited for mammography screening in Norway	Whether a decision was based on information provided in the national leaflet for mammography screening	Grateful for invitation, difficult to cope with the fear of potential disease, responsible health behaviour, worry avoidance	Assumption 1: The women did not base their decision on information in the leaflet and felt that it was no decision to participate or not.
Henriksen et al. 2015 [24]	Interview study that explores the influence of framing in information material in breast cancer screening in Denmark	Decision-making processes were not based on the information that accompanied the screening invitation	Attitudes from family and friends are important, do not need information about harms, screening is important, accepts harms, leaflets are instructions	Assumption 2: The women misinterpreted the harm. Assumption 3: Participation is a moral obligation and decision is influenced by family.
Byskov Petersen et al. 2020 [25]	Interview study on the understandability of benefits and harms of cervical cancer screening in Denmark	Adequate information plays a central role in the process of informed choice	Numbers don’t matter the lives do, unaware and shocked about harm, misinterpret and understate harms	Assumption 2: The women downplayed or ignored the harms.
Gram et al. 2023 [26]	Interview study on the experiences of being discontinued from mammography screening in Denmark	Whether women are aware of why they are being discontinued from screening	Societal value of invitation, participation is a moral obligation, gratitude	Assumption 2: Women exempted themselves from the risk of being harmed. Assumption 3: Participation is a moral obligation and is considered authoritative.
Jensen et al. 2021 [27]	Deliberative poll using video information material on mammography screening in Denmark	Are opinions about screening consistent with knowledge about screening	Information increases knowledge about screening but does not change opinions or willingness to participate	Assumption 2: The women altered their preferences for screening to fit the available information.
Lindberg et al. 2013 [28]	Interview study on the long-term experience of mammography screening in Denmark	Communication regulates choices and forms these as right or wrong	Fear of cancer, gratitude, confidence in screening, seeking confirmation of well-being	Assumption 3: Not reluctant to participate, grateful for the screening program

## Data Availability

Data is unavailable due to privacy and ethical restrictions.

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
