# Peer review of "Questioning ‘Informed Choice’ in Medical Screening: The Role of Neoliberal Rhetoric, Culture, and Social Context"

_healthcare, 2023, doi:10.3390/healthcare11091230_

Round 1
Reviewer 1 Report
It is considered to be a rare study that performed a qualitative analysis on the choice of medical screening.
2.1. Methods and materials
"We extracted data from peer-reviewed articles and discussed it within the author group."
It would be nice if you could explain the process of peer-review and discussion within the author group in a bit more detail.
It would be better if some sentences could be edited to be suitable for academic paper.
Author Response
Thank you for your valuable input and your time reviewing our article. We have elaborated on the methodology and we rewrote the entire section with the help of a native colleague.
Reviewer 2 Report
Despite the subject under study, it is unusual to analyze a paper where there is no figure, table, or scheme to illustrate/support the content of the document.
The applied methodology should be present in the abstract, as well as the contributions/findings.
The application of the snowball methodology is not explicit throughout the document. It should be clear which studies constituted the initial set, the structure, and the inclusion/exclusion criteria that allowed iteration of the method, as well as reaching the saturation point.
All these considerations lead to a set of limited conclusions where further work resulting from the study is not observed.
Author Response
Thank you for your valuable input and your time reviewing our article. We have added a sentence about the methodology to the abstract. We also added, to the method section (2.1 methods and materials), how we used the snowball methodology.
We have followed your suggestion and added a table where we present the articles included in the analysis, their study design, and their answer to sub-question one and two that informed the analysis (2.1 Methods and materials)
Reviewer 3 Report
The paper touches on an important issue in healthcare, challenging the commenly accepted concept of informed choice/consent.
However, it appears to be the case that the paper investigate the issue in a specific contenxt of Sweden. But this perspective is not clear enough in my opinion. Otherwise, the discussion of informed choice seems to be echoing a well-established tradition of rethinking informed consent.
To conclude, it is an interesting piece and I enjoy reading. And I would like to see more discussion based in Swedish context i.e. how it is different to other contexts.
Author Response
Thank you for your valuable input and your time reviewing our article. Thank you for your interest in the discussion of the informed choice. We have now related our findings to other contexts in the discussion under the section 4.2.
Reviewer 4 Report
For purposes of clarity, it would be useful to tabulate some of the results. This gives the reader information about how many articles were reviewed and what these articles found/concluded. It would also be helpful to include a line or two about how these articles were identified and included/excluded for discussion.
Author Response
Thank you for your valuable input and your time reviewing our article. Another reviewer also suggested that we include more information about how we retrieved the articles, e.g. the snowball method, therefore, we have added that to the method section (2.1 methods and materials).
We have followed your suggestion and added a table where we present the articles included in the analysis, their study design, and their relation and support for the analysis/results (sub-questions that inform the analysis).
Round 2
Reviewer 2 Report
Some improvements were found in the abstract, the introduction, as well as throughout the document.
However, the authors state that:
“…We have followed your suggestion and added a table where we present the articles included in the analysis, their study design, and their answer to sub-question one and two that informed the analysis (2.1 Methods and materials)”
I could not find the referred table.
I also suggested improving the conclusion section, but I could not find any change.
Author Response
Thank you for once again reviewing our manuscript, we appreciate your time and effort. We are sorry that the table was not properly available to you; we have now placed the table within the text and introduced it at the beginning of the results section.
We have also revised the conclusion in regard to language and preciseness. We hope that you find the changes adequate.